# The Discovery of Oropharyngeal Microbiota with Inhibitory Activity against Pathogenic *Neisseria gonorrhoeae* and *Neisseria meningitidis*: An In Vitro Study of Clinical Isolates

**DOI:** 10.3390/microorganisms10122497

**Published:** 2022-12-16

**Authors:** Elvis Achondou Akomoneh, Jolein Gyonne Elise Laumen, Saïd Abdellati, Christophe Van Dijck, Thibau Vanbaelen, Xavier Basil Britto, Sheeba S. Manoharan-Basil, Chris Kenyon

**Affiliations:** 1HIV/STI Unit, Department of Clinical Sciences, Institute of Tropical Medicine, Nationalestraat 155, 2000 Antwerp, Belgium; 2Department of Microbiology and Parasitology, University of Bamenda, Bambili P.O. Box 39, Cameroon; 3Laboratory of Medical Microbiology, University of Antwerp, 2610 Wilrijk, Belgium; 4Clinical Reference Laboratory, Department of Clinical Sciences, Institute of Tropical Medicine, 2000 Antwerp, Belgium; 5Division of Infectious Diseases and HIV Medicine, University of Cape Town, Anzio Road, Observatory 7700, South Africa

**Keywords:** Microbiome, colonisation resistance, *Neisseria gonorrhoeae*, *Neisseria meningitidis*, Listerine mouthwash

## Abstract

With increasing incidence of pathogenic *Neisseria* infections coupled with emerging resistance to antimicrobials, alternative approaches to limit the spread are sought. We investigated the inhibitory effect of oropharyngeal microbiota on the growth of *N. gonorrhoeae* and *N. meningitidis* and the impact of the essential oil-based mouthwash Listerine Cool Mint^®^ (Listerine). Oropharyngeal swabs from 64 men who have sex with men (*n* = 118) from a previous study (PReGo study) were analysed (ClinicalTrials.gov, NCT03881007). These included 64 baseline and 54 samples following three months of daily use of Listerine. Inhibition was confirmed by agar overlay assay, and inhibitory bacteria isolated using replica plating and identified using MALDI-TOF. The number of inhibitory isolates were compared before and after Listerine use. Thirty-one pharyngeal samples (26%) showed inhibitory activity against *N. gonorrhoeae* and/or *N. meningitidis*, and 62 inhibitory isolates were characterised. Fourteen species belonging to the genera *Streptococci* and *Rothia* were identified. More inhibitory isolates were observed following Listerine use compared to baseline, although this effect was not statistically significant (*p* = 0.073). This study isolated and identified inhibitory bacteria against pathogenic *Neisseria* spp. and established that daily Listerine use did not decrease their prevalence. These findings could provide a new approach for the prevention and treatment of pharyngeal *Neisseria* infections.

## 1. Introduction

The genus *Neisseria* includes two human pathogenic species, *N. gonorrhoeae* and *N. meningitidis* [1]. *Neisseria meningitidis* is a coloniser of the nasopharynx from where it can become invasive, causing sepsis and cerebrospinal meningitis [2]. *Neisseria gonorrhoeae*, on the other hand, causes infections in the urinogenital tract, rectum, and pharynx [3]. *N. gonorrhoeae* infections can also result in neonatal conjunctivitis if untreated women transmit the pathogen during delivery [4]. Over the years, the number of infections with *N. gonorrhoeae* have been on the rise [5]. Unfortunately, the number of available antibiotics is on the decline due to the emergence of resistance with some gonococcal isolates identified with resistance to all recommended antimicrobials [6,7].

Considering the alarming rate at which gonococcal isolates are developing resistance against conventional antibiotics, the search for novel antimicrobials from other sources, including natural sources, is increasingly gaining attention [8,9]. Screening the human microbiota for bacteria with inhibitory effects against the pathogen for use as probiotics or the production of bacteriocins could be a credible alternative [10]. Bacteriocins are bactericidal proteins or peptides secreted by a particular species of bacteria and act against other closely related species but not against the producing strain [11]. Compared with conventional antibiotics, bacteriocins may have the advantages of safety, high efficacy, stability, and less susceptibility to resistance-induction [12]. They have been shown to prevent infection and play key roles in regulating host immune and inflammatory responses [10]. Of particular relevance to *N. gonorrhoeae*, bacteriocins could offer a new avenue to combating antibiotic resistant strains [13]. To the best of our knowledge, no study has screened the oral flora for the presence of organisms with an inhibitory effect against pathogenic *Neisseria* species [14,15].

In this study, we screened for the presence of such inhibitory bacteria in samples from a randomized controlled trial we performed to investigate the efficacy of another antibiotic sparing agent (oral rinsing with Listerine), to reduce the incidence of bacterial sexually transmitted infections (STIs) in men who have sex with men (MSM) (the PReG0 study) [16]. In-vitro studies and pilot in-vivo studies revealed that the essential oil-based mouthwash Listerine Cool Mint^®^ (herein after referred to as Listerine) (Johnson & Johnson, New Brunswick, NJ, USA), was bactericidal against *N. gonorrhoeae* and other bacterial species [17]. Surprisingly, this study found that the use of Listerine resulted in a small but statistically significant increase in the incidence of oropharyngeal *N. gonorrhoeae* [16]. Chow et al. [18], in another study (the OMEGA study), reported similar trends, although without establishing a statistically significant increase in the cumulative incidence of oropharyngeal gonorrhoea between MSM who used Listerine Zero^®^ versus Biotène. We hypothesised that this effect might have been mediated by Listerine reducing the abundance of commensal bacteria responsible for colonization resistance. It is worth noting that, indigenous human microbiota use a number of modalities, such as the secretion of bacteriocins, and competition for space and resources to resist the colonisation and invasion of the host by exogenous bacteria (termed colonisation resistance) [19]. The effectiveness of colonisation resistance relies on the presence of healthy human microbiota at the possible site of colonisation and changes in the microflora structure could lead to a shift in the balance in advantage to a pathogenic species composition [20]. The use of broad-spectrum antibiotics, for instance, could diminish key commensal species, leaving a void that could be rapidly filled by pathogens [21]. Listerine may have a similar effect [17].

Despite these findings, Listerine and other antiseptic mouthwashes have been proposed and are being used as an intervention to reduce gonorrhoea transmission in MSM [22,23]. There is limited information on the overall long-term effect of these mouthwashes on the oral microbiome and colonisation resistance. Establishing this is key to explaining the findings of the OMEGA and PReGo clinical trials and other prospective studies that employed the long-term use of Listerine [16,24]. Moreover, some in vitro studies have shown that oil-based antiseptic mouthwashes are bactericidal against a range of species through the disruption of cell walls and inactivation of essential enzymes [25]. In addition, some of these mouthwashes contain high concentrations of alcohol which has been linked to a shift in the balance of the oral microbiome [26,27]. Any alteration that negatively affects the oral microbiome could possibly act against colonisation resistance and increase the individual’s susceptibility to infections. In this study, we aimed firstly, to identify the prevalence of oropharyngeal inhibitors of *N. gonorrhoeae* and *N. meningitidis* in the participants of the PreGo study and secondly, to determine if the use of Listerine has an effect on the prevalence and number of these inhibitory bacteria.

## 2. Materials and Methods

### 2.1. Participants and Specimens

This sub study analysed samples from the Preventing Resistance to Gonorrhoeae (PReGo) study (ClinicalTrials.gov, NCT03881007) [16]. This was a randomised, double-blind, placebo-controlled single centre trial that assessed the superiority of Listerine over a placebo mouthwash in preventing bacterial STIs among 64 MSM using HIV pre-exposure prophylaxis (PrEP). Participants were randomly assigned to first receive Listerine followed by placebo for 3 months each (Listerine-placebo arm) or a placebo mouthwash followed by Listerine for 3 months each (placebo-Listerine arm). Samples analysed in this study included 118 oropharyngeal swabs collected from the 64 participants before the use of Listerine (baseline; 64 samples) and following 3 months of daily use of Listerine (treatment; 54 samples). The samples collected following the use of placebo were not analysed.

This study being a continuation of the PReGo study was also covered by the ethical approvals from the Institutional Review Board of the Institute of Tropical Medicine, Antwerp (Ref: 1276/18) and the Ethics Committee of the University of Antwerp (Ref: 19/06/058).

### 2.2. Origin of Pathogenic Neisseria Test Species

This study tested the inhibition of the growth of two pathogenic *Neisseria* species, *N. gonorrhoeae* WHO P, a WHO reference strain [28] and *N. meningitidis* M00003/, a clinical isolate, obtained from an oropharyngeal swab from an asymptomatic man participating in the Resistogenicity Study at ITM [29]. Some of the isolates are resistant to a number of recommended antibiotics for treatment (Table 1).

### 2.3. Identification of Inhibitory Commensals

The 118 oropharyngeal swabs analysed were previously cultured on Columbia blood agar plates and stored at –80 °C in skimmed milk. A loopful of each sample was diluted (1:100,000) and inoculated on Columbia Nalidixic Acid Agar (CNA) plates selective for gram positive cocci to obtain single colonies. Replica plating of each subculture was made on GC agar plates following the method of Lederberg and Lederberg [30]. Specifically, each CNA agar plate was replicated by stamping on a sterile piece of velvet pads and exerting only slight pressure. The CNA plate was removed, and three GC agar plates were pressed gently against the velvet pads in turns. All three replica plates were incubated overnight at 37 °C with 5% CO_2_. Growth inhibition of the *N. gonorrhoeae* and *N. meningitidis* strains was assessed by agar overlay assay as previously described [31,32]. Briefly, 100 µL suspension of *N. gonorrhoeae*/*N. meningitidis*, in phosphate-buffered saline (PBS) approximately 10^6^ CFU/mL, was diluted in 10 mL of melted GCB agar supplemented with 100 µL IsoVitaleX (Oxoid^TM^, Hampshire, England). This was overlayed (in turns) on two sets of the triplicate replicas and further incubated for 24 to 48 h. Following incubation, inhibitory colonies were observed and isolated from the third set of replicas. Each inhibitory colony that was obtained was purified on Columbia blood agar and reconfirmation of the isolates at the species level was performed by MALDI Biotyper IVD (Bruker Daltonics, Bremen, Germany) (library updated to v.10.0.0.0_8326-9468 (IVD).

The number of inhibitory colonies were compared between baseline and samples following three months of daily Listerine use. Due to the cross-over study design, the ‘month 3′ specimens could be from the month 3 or 6 visits but in all cases, they were specimens that were taken after 3 months exposure to Listerine. Results were presented as counts and percentages of samples positive. The paired samples t-test was used to compare the mean number of inhibitory colonies before and after the receipt of Listerine at a 95 % significant level. The statistical analyses were performed in STATA MP v.16 (StataCorp., College Station, TX, USA).

## 3. Results

### 3.1. Isolation and Identification of Inhibitory Commensal Bacteria

We analysed oropharyngeal swabs to identify commensal bacteria with inhibitory activity against two pathogenic *Neisseria* species, *N. gonorrhoeae* WHO P and *N. meningitidis* M00003/1. Our screening was restricted to gram positive commensal bacteria as the initial growth of the samples was on CNA agar, a selective medium for this group. Of the 118 samples, 31 (26%) had inhibitory colonies, while 87 (74%) had none. Eight of the 31 samples had only one inhibitory colony, while 26 had multiple inhibitory colonies (Figure 1). A total of 62 inhibitory colonies were isolated from the 118 samples (Appendix A).

Furthermore, we wanted to elucidate whether the 62 isolates were inhibitory against *N. gonorrhoeae* or *N. meningitidis* or both pathogenic *Neisseria* species. Fifty-four isolates were inhibitory to *N. gonorrhoeae*, 41 to *N. meningitidis* and 33 were inhibitory to both pathogenic *Neisseria* species (Figure 2).

All 62 inhibitory isolates were identified by MALDI-TOF MS and they belonged to the genus *Streptococci* (12 species) and genus *Rothia* (2 species) (Table 2). The most prevalent species were *S. parasanguinis* (14; 22.6%), *S. sanguinis* (10; 16.1%) and *S. mitis* (8; 12.9%). One isolate each belonging to *R. mucilaginosa*, *S. gordonii*, *S. infantis* and *S. peroris* were identified. It is worth noting that more than one inhibitory species was identified in the same participant in some cases (Appendix A).

### 3.2. Comparing the Number of Inhibitory Isolates before and after Receipt of Listerine

The number and prevalence of inhibitory isolates in the baseline samples versus samples following three months of use of Listerine (treatment) were assessed in this study. Surprisingly, more inhibitory isolates were in samples after Listerine treatment (20/54, 37%) as compared to baseline samples (11/64, 17.2%) (Appendix A), although the difference was not statistically significant (*p* = 0.073, 95% CI −1.2–0.5). Likewise, the number of inhibitory species were greater in samples following Listerine treatment (13 species) as compared to baseline samples (9 species). Notably, *R. mucilaginosa* was the only isolate that was present in the baseline samples but absent in the treatment samples. The two *R. dentocariosa* isolates and *S. gordonii*, *S. infantis*, *S. peroris* and *S. vestibularis* were present only in the post-Listerine samples (Table 3). It is worth noting that, inhibitory species were isolated in some participants at baseline but not after treatment, while in others these were identified after treatment but not in baseline samples (Appendix A).

We also noted the number of inhibitory colonies in participants who tested positive for pharyngeal gonorrhoea by PCR in the PReGo study (Appendix A). In total, four participants tested positive for pharyngeal gonorrhoea, three at baseline and one following three months of Listerine use. Two of the three positive cases at baseline tested negative after three months of Listerine use. We identified at least three inhibitory bacteria isolates from each of these participants. The third participant dropped out of the study. On the other hand, the only participant who tested positive for pharyngeal gonorrhoea after three months of Listerine use had previously tested negative at bassline. One inhibitory isolate was identified from sample from this participant.

## 4. Discussion

The high levels of antimicrobial resistance in pathogenic *Neisseria* species provide strong motivation to search for novel approaches to prevent and treat these infections [6,7]. This study screened for oropharyngeal gram-positive cocci with inhibitory effects on pathogenic *Neisseria* species. The primary aim was to identify commensal bacteria as putative candidates for possible use, either as probiotics in preventing pathogenic antibiotic resistant *Neisseria* colonisation of the human host or their secretory products as therapeutics. We identified 14 different species of *Streptococci* and *Rothia* from 31 individuals capable of inhibiting the growth of *N. gonorrhoeae* and/or *N. meningitidis* in vitro. More interesting is the fact that 13 of the 14 species are non-pathogenic bacteria, with some even designated by the Food and Drug Administration (FDA) with the GRAS (generally regarded as safe) status by their virtuous nature [33] or listed in The International Dairy Federation to have a safe history [34], indicating that they can be used without any demonstrable harm to consumers. These findings add to other studies exploiting colonisation resistance in combating pathogenic bacteria but, more specifically, the hopes of containing multidrug resistant *N. gonorrhoeae* [32,35]. Worth noting is the fact that, pharyngeal gonorrhoea could serve as an important reservoir and source of urethral gonorrhea as it is more difficult to eradicate and requires a different treatment strategy. This also makes it a potential source of drug resistant *N. gonorrheae* [36].

Previous studies have found that a range of bacteria inhibit the growth of *N. gonorrhoeae*. These include other *Neisseria* spp. [15,37] as well as the *Lactobacilli*, *Staphylococci*, *Streptococci*, and *Escherichia* which are part of the human microbiome [14,31,38,39]. Exploring the mechanisms employed by these bacteria is important as their coexistence within the human host offers the possibility of a therapeutic agent with minimal effect on the host and the microbiome. Importantly, if bacteriocins are identified as the main interference mechanism employed by the species, this could provide a credible alternative to antibiotics in treating multidrug-resistant Neisseria gonorrhoeae infections. This, owing to the fact that bacteriocins identified against other pathogenic bacteria have displayed more efficacy, tolerance and less susceptibility for resistance to develop against them [40].

A secondary aim of this study was to evaluate if Listerine mouthwash reduced the prevalence of bacteria with inhibitory effects against pathogenic *Neisseria* spp. Listerine Cool Mint^Ò^ is a 22% hydroalcoholic solution containing menthol, thymol, methyl salicylate and eucalyptol. This or other antiseptic solutions are commonly used by high-risk populations such as MSM on HIV PrEP to prevent STIs, including gonorrhoea [41,42]. Unfortunately, there is a possibility of the antiseptics altering the composition of the oral microbiome. We observed an increase in the number of inhibitory bacteria following three months of use of Listerine compared to baseline. Although this effect was not statistically significant and may be a chance finding, a number of participants were found to have more inhibitory bacteria after three months of Listerine use than at baseline, corroborating other findings [16,24,43]. We also observed that participants who tested positive for pharyngeal gonorrhoea at baseline tested negative after three months of Listerine use with at least three inhibitory isolates identified from each of these participants. Moreover, metagenomic analysis of these swabs revealed a significant change in the composition of the oral microbiome after the use of Listerine in favor of *Streptococcus anginosus* and *Fusobacterium nucleatum* [44]. Other studies have found an association between the relative abundance of these bacteria and alcohol intake [26,27]. The high alcohol concentration in Listerine Cool Mint^®^ (22%) may therefore explain some of our findings. It is important to note that, maintaining a healthy microbial community in the oropharynx is important not only to prevent colonisation and infection of the host, but also in various metabolic and immunological processes [45]. We were unable to investigate any of these effects in our study.

The identification of human microbiota with inhibitory activity against a multidrug resistant *N. gonorrhoeae* presents the possibility of using these commensal isolates as probiotics [46]. If bacteriocins are confirmed as the main interference mechanism from these isolates, then these could be developed to provide a novel treatment for multidrug-resistant *N. gonorrhoeae* infections [40].

We recommend further research should be undertaken to investigate the interference mechanism of the 14 species of bacteria to identify and characterise all plausible bacteriocins, determine their amino acid residues, bioactivity in vivo, minimum inhibitory, bactericidal concentrations, tolerance, absorption, distribution, elimination and susceptibility to resistance-induction. A limitation in the study was the use of selective media for gram positive bacteria and serial dilution of the sample which might have resulted in exclusion of other inhibitory bacteria with therapeutic potentials. We recommend plating of multiple dilutions on nonselective media to increase the possibility of isolating additional inhibitory bacteria. We also recommend testing the inhibitory isolates against different *N. gonorrhoeae*, *N. meningitidis* and commensal *Neisseria* spp. strains including clinical strains currently in circulation in order to have a more comprehensive overview of the spectrum of strains inhibited.

## 5. Conclusions

This study successfully isolated and identified inhibitory bacteria against pathogenic *Neisseria* spp. from the human oropharyngeal microbiome. We also established that long-term use of the essential oil-based mouthwash Listerine does not reduce the prevalence of oropharyngeal commensals with inhibitory activity against *N. gonorrhoeae* and *N. meningitidis*.

## Figures and Tables

**Figure 1 microorganisms-10-02497-f001:**
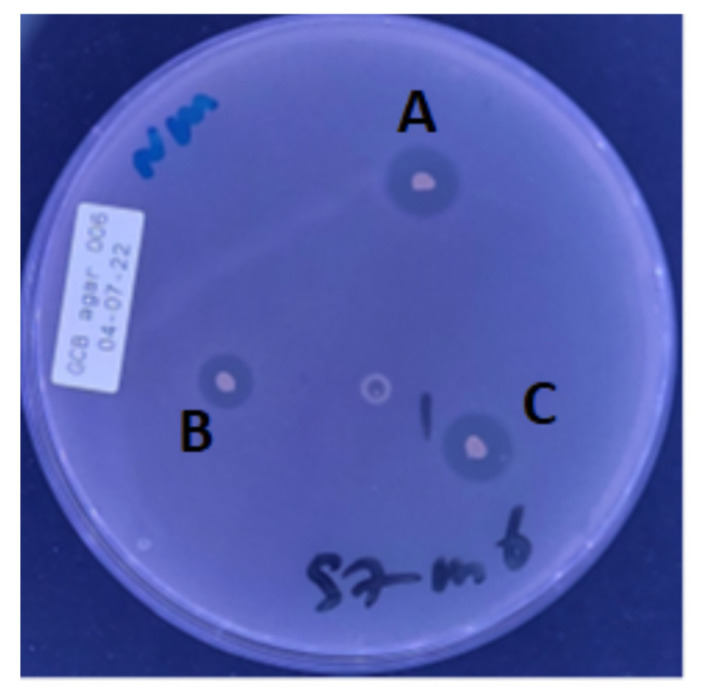
Agar overlay assay showing inhibitory activity by some isolates to *N. meningitidis* M00003/1: A, *Rothia dentocariosa* B, *Streptococcus mitis *and C, *Rothia dentocariosa*.

**Figure 2 microorganisms-10-02497-f002:**
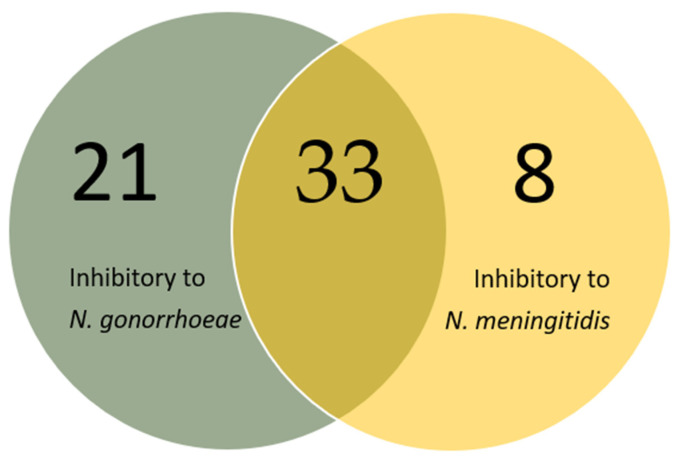
The number of isolates found to be inhibitory to two pathogenic *Neisseria* species.

**Table 1 microorganisms-10-02497-t001:** Susceptibility values of test strains.

Species	Strain	AZM MIC	AZM	CRO MIC	CRO	Source
*N. gonorrhoeae*	WHO P	4	R	0.004	S	Unemo et al. (2016) [28]
*N. meningitidis*	MO0003/1	1	S	<0.016	S	de Block et al. (2021) [29]

Minimum Inhibitory Concentrations (MIC), Azithromycin (AZM), Ceftriaxone (CRO), Susceptible (S), Resistant (R).

**Table 2 microorganisms-10-02497-t002:** Inhibitory isolates against pathogenic *Neisseria* species.

Isolate	Frequency	Percent
*Rothia dentocariosa*	2	3.2
*Rothia mucilaginosa*	1	1.6
*Streptococcus cristatus*	4	6.5
*Streptococcus dysgalactiae*	3	4.8
*Streptococcus gordonii*	1	1.6
*Streptococcus infantis*	1	1.6
*Streptococcus mitis*	8	12.9
*Streptococcus oralis*	5	8.1
*Streptococcus parasanguinis*	14	22.6
*Streptococcus peroris*	1	1.6
*Streptococcus pyogenes*	6	9.7
*Streptococcus salivarius*	4	6.5
*Streptococcus sanguinis*	10	16.1
*Streptococcus vestibularis*	2	3.2
Total	62	100

**Table 3 microorganisms-10-02497-t003:** Number of inhibitory isolates against pathogenic Neisseria species before and after treatment.

Isolate	Frequency in Baseline	Frequency in Treatment
*Rothia mucilaginosa*	1	0
*Rothia dentocariosa*	0	2
*Streptococcus cristatus*	3	1
*Streptococcus dysgalactiae*	2	1
*Streptococcus gordonii*	0	1
*Streptococcus infantis*	0	1
*Streptococcus mitis*	4	4
*Streptococcus oralis*	2	3
*Streptococcus parasanguinis*	3	11
*Streptococcus peroris*	0	1
*Streptococcus pyogenes*	1	5
*Streptococcus salivarius*	3	1
*Streptococcus sanguinis*	4	6
*Streptococcus* *vestibularis*	0	2

## Data Availability

The data supporting the findings of this study are retained at ITM and because of ethical and privacy concerns will not be made openly accessible. ITM adheres to the FAIR data principles (findable, accessible, interoperable, and reusable) and recognises that data should be “as open as possible and as closed as necessary”. Anonymised, individual participant data of the study as well as additional related documents, such as the study protocol, the annotated case report form, the data dictionary, and statistical analysis scripts can be made available within 12 months after the publication of the study results and without end date. Data will be retained at the ITM data repository and can be requested via a mail to ITM’s central point for research data access at ITMresearchdataaccess@itg.be, (accessed on 6 November 2022). A governed data access mechanism applies including (1) completion of data request form, (2) evaluation by a data access committee, (3) signing of a data sharing agreement, and (4) secure transfer of data.

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
