# Peer review of "The Discovery of Oropharyngeal Microbiota with Inhibitory Activity against Pathogenic Neisseria gonorrhoeae and Neisseria meningitidis: An In Vitro Study of Clinical Isolates"

_microorganisms, 2022, doi:10.3390/microorganisms10122497_

Round 1

Reviewer 1 Report

Thank you for asking me to review this paper.

The project uses already collected samples from a trial of mouthwash to prevent oropharyngeal gonorrhoea.  The authors are assessing the Inhibitory bacteria that are in the oropharyngeal microbiota and their effect on the growth of N. gonorrhoeae and N. meningitidis  before and after the use of mouthwash.

The paper is well written and apart from some obvious adjustments to the figures where the numbers are not aligned I have no significant changes to the paper. Some issues that the authors might like to consider is including some discussion about whether the magnitude of the inhibition provided by these bacteria could conceivably have a clinically significant effect on the colonisation in the throat. I note on the plate that was provided in the figure it was not marked.

One issue that would be a benefit to the introduction or to the discussion, is to outline for the readers why it is that oral pharyngeal gonorrhoea is so important in the transmission of infection. I'm familiar with a debate in Lancet infectious diseases where the arguments for this I described.

Reviewer 2 Report

Major comment

The study describes oropharyngeal microbiota with inhibitory activity against pathogenic Neisseria gonorrhoeae.  The study/manuscript could be strengthened by looking at the prevalence of these commensal bacteria in individuals who had pharyngeal NG infections during the study.

Minor comments

For clarity, please add pharyngeal to “Thirty-one samples (26%)” in the abstract.

Lines 31 and 32 - “These findings could provide a new approach for the prevention and treatment of Neisseria infections”.  Authors should add pharyngeal infections.   

Line 40 – please consider replacing the term “inhabitant” with a more suitable term.   Inhabitant suggest that it is NG commonly inhabitants those anatomical sites.

Why was the WHO-P strain selected from this study?

Lines 123 – 127.  The following description is confusing. “The 118 oropharyngeal swabs (ESwabTM COPAN Diagnostics Inc., Brescia, Italy) analysed were previously cultured on Columbia blood agar plates and stored at –80 oC in skimmed milk. A loopful of each sample was diluted (1:100,000) and inoculated on Columbia Nalidixic Acid Agar (CNA) plates selective for gram positive cocci to obtain single colonies.”  Was the sample that was diluted taken directly from the ESwabTM media or from the previous cultures?

Do the authors have any concerns that the “inhibitory” strains were more resistant to freeze/thaw cycles and are therefore more represented in this study?

Were the authors able to perform any analysis comparing samples from participants who tested positive for NG during study to those who were NG-negative?  This is not described in the material and methods section.

Figure 2 needs to be fixed as the numbers do not match to those described in the text.  For example, the figure shows that only 8 were inhibitory to N. meningitidis.

The authors should state that only gram-positive bacteria were studied as a limitation of the study.

Reviewer 3 Report

This is a lab study to identify oropharygeal organisms collected from high risk individuals which inhibit  growth of Neisseria species in vitro. 

Major comments: The study and results are fairly straightforward but usefulness is limited by the lack of in vivo correlation. It may be useful to identify inhibitory compounds released by these organisms but interactions in culture are a poor substitute for the interactions that occur in vivo, so usefulness is limited. It seems that the authors had access to the metagenomic analyses, was there any correlation between prevalence of these inhibitory organisms and finding of Neisseria species in those subjects? That finding, if present, would greatly improve the significance of the study. 

Minor comments

There appear to be sections of the text which are in different font size. This should be edited prior to publication. 
